# Exploring Volatile Organic Compounds in Rhizomes and Leaves of *Kaempferia parviflora* Wall. Ex Baker Using HS-SPME and GC–TOF/MS Combined with Multivariate Analysis

**DOI:** 10.3390/metabo13050651

**Published:** 2023-05-11

**Authors:** May San Thawtar, Miyako Kusano, Li Yingtao, Min San Thein, Keisuke Tanaka, Marlon Rivera, Miao Shi, Kazuo N. Watanabe

**Affiliations:** 1Degree Programs in Life and Earth Sciences, Graduate School of Science and Technology, University of Tsukuba, Tsukuba 305-8572, Japan; s2036030@u.tsukuba.ac.jp (M.S.T.);; 2Tsukuba-Plant Innovation Research Center, Institute of Life and Environmental Sciences, University of Tsukuba, Tsukuba 305-8572, Japan; 3Postharvest Research Institute, Ministry of Agriculture, Livestock, and Irrigation, Yezin, Myanmar; 4Department of Agricultural Research, Ministry of Agriculture, Livestock, and Irrigation, Yezin, Myanmar; 5NODAI Genome Research Center, Tokyo University of Agriculture, Setagaya 156-8502, Japan; 6Faculty of Informatics, Tokyo University of Information Sciences, Chiba 65-8501, Japan; 7Institute of Biological Sciences, University of the Philippines Los Baños, Laguna, Philippines

**Keywords:** *Kaempferia parviflora*, volatile organic compounds (VOCs), growth room, field, headspace solid-phase microextraction (HS-SPME), gas chromatography/time-of-flight mass spectrometry (GC-TOF-MS)

## Abstract

Volatile organic compounds (VOCs) play an important role in the biological activities of the medicinal Zingiberaceae species. In commercial preparations of VOCs from *Kaempferia parviflora* rhizomes, its leaves are wasted as by-products. The foliage could be an alternative source to rhizome, but its VOCs composition has not been explored previously. In this study, the VOCs in the leaves and rhizomes of *K. parviflora* plants grown in a growth room and in the field were analyzed using the headspace solid-phase microextraction (HS-SPME) method coupled with gas chromatography and time-of-flight mass spectrometry (GC-TOF-MS). The results showed a total of 75 and 78 VOCs identified from the leaves and rhizomes, respectively, of plants grown in the growth room. In the field samples, 96 VOCs were detected from the leaves and 98 from the rhizomes. These numbers are higher compared to the previous reports, which can be attributed to the analytical techniques used. It was also observed that monoterpenes were dominant in leaves, whereas sesquiterpenes were more abundant in rhizomes. Principal component analysis (PCA) revealed significantly higher abundance and diversity of VOCs in plants grown in the field than in the growth room. A high level of similarity of identified VOCs between the two tissues was also observed, as they shared 68 and 94 VOCs in the growth room and field samples, respectively. The difference lies in the relative abundance of VOCs, as most of them are abundant in rhizomes. Overall, the current study showed that the leaves of *K. parviflora*, grown in any growth conditions, can be further utilized as an alternative source of VOCs for rhizomes.

## 1. Introduction

Volatile organic compounds (VOCs) are one of the most important secondary metabolites produced by plants. They are involved in the vital physiological processes of plants, including pollinator attraction, plant–plant interaction, and as defense mechanisms against insect and herbivore attacks [1,2]. VOCs are synthesized from different plant organs, such as the leaves, flowers, rhizomes, and roots [3,4]. Their emission from each organ is strongly influenced by light, temperature, CO_2_, and genetics, as well as in response to abiotic and biotic stresses [5,6]. In herbal plants, VOCs are mainly essential oil components, and are responsible for their aroma, flavor, and pharmaceutical properties [7,8,9]. The major classes of VOCs emitted from most aromatic plants are mainly terpenes (monoterpenes, sesquiterpenes, and their derivatives), oxygen derivatives (alcohols, aldehydes, ketones, and esters) and fatty acid derivates [10,11,12]. Around 85–90% of the world’s population rely on herbal products in their daily lives as medicine, food, and cosmetics [13]. Changes in the composition of VOCs can affect the bioactivity of aromatic herbs, which in turn alter the quality of its products [14]. In addition, studies exploring VOCs from underexploited herbal plants have increased in order to develop novel medicines for pandemic diseases occurring nowadays [15,16].

*Kaempferia parviflora* Wall. ex Baker is one of the underexploited medicinal species under the family Zingiberaceae. It is famous for its deep, purple-colored rhizome and is commonly known as ‘black ginger’ [15,17]. It is widely distributed throughout Southeast Asia, where it has different local names: Thailand (Kra-Chai-Dam), Myanmar (Nannwin Nat), and Malaysia (Cekur hitam) [18,19]. It can also be found in other Asian countries such as India and China [20,21]. This rhizome has been used in folk medicine for many centuries to cure erectile dysfunction, gastrointestinal complaints, high blood glucose, and metabolic disorders [22,23,24,25], and for the improvement of endurance, vigor, and muscle strength [26]. A variety of pharmacological activities has also been reported from its extracts, such as anti-inflammatory [27], antioxidant [28,29], antiallergic [30], anticancer [31,32], anticholinesterase [33], antimicrobial [22], anti-obesity [34], neuro-protective [35,36,37,38], sexually enhancing [39], anti-acne [40], and antimutagenic [41] properties.

Most of the phytochemical studies on *K. parviflora* in the 2020s have discussed the isolation and structural identification of methoxy flavones in rhizomes and the evaluation of its pharmaceutical properties [30,33,41,42,43,44]. In addition, the presence of phenolic glycosides [45,46], acetophenone [47], and terpenoids [15,48,49,50] has been reported from the rhizome extracts. On the other hand, the volatile compounds present in rhizomes are poorly studied; there are only five studies available to date [15,48,49,50,51], and different results regarding the quantities and concentrations of the identified compounds were reported. The essential oils of the plant also remained unexplored for their pharmaceutical applications [15]. It is, therefore, of great interest to better understand the complete VOC profile of the rhizomes and explore more of its value.

Due to the recognition of bioactive properties of *K. parviflora* rhizome extracts, they has been extensively used in the pharmaceutical, food, cosmetics, and fragrance industries. Its mass collection to meet the growing demand led to the depletion of wild resources, which affected the natural population of *K. parviflora* [52]. Moreover, the availability of rhizomes is becoming limited because of its long dormancy period; it usually takes a 12-month growing period to reach maturity [53,54,55]. These factors led to the scarcity of sources to fulfill the need for economic purposes.

In recent decades, there has been a growing interest in exploring non-target, agricultural by-products as potential sources of industrially important compounds. In other *Kaempferia* species, such as *K. galanga*, the foliage contains several bioactive metabolites [56,57] and has shown sedative [58], anti-inflammatory [59], and antioxidant properties [60]. It is also used in food flavorings, beverages, and cosmetic product preparations [60,61,62]. In *K. parviflora*, after harvesting the rhizomes, the leaves are discarded, and their potential uses are poorly explored. To date, only two studies on the screening of bioactive metabolites from foliage are available. Park et al. (2021) investigated a total of 42 compounds, including 26 flavonoids and 16 phenolics, from the leaf extracts of in vitro- and ex vitro-grown plants using liquid chromatography–mass spectrophotometry (LC-MS/MS) [63]. The results also showed that some of the methoxy flavones present in rhizomes were identified in the leaves. In addition, the lipophilic metabolite profile in the leaf extracts was explored by Song et al. (2021) using liquid chromatography–multiple reaction monitoring–mass spectrophotometry (LC–MRM–MS) and reported the presence of carotenoids, tocopherols, fatty acids, phylloquinone (Vitamin K1), and phytosterols [64]. Other than those compounds, the profile and content of VOCs in *K. parviflora* leaves have not been examined, but should also be explored. It is interesting to investigate whether the leaves can be a potential alternative source of economically important volatiles for rhizomes, considering the scarcity of rhizomes in the present-day market.

In the characterization of VOCs from plant matrices, the sample preparation and detection techniques are critical steps in achieving optimal results [65,66]. Nowadays, headspace solid-phase microextraction (HS-SPME) sample preparation coupled with gas chromatography/time-of-flight mass spectrometry (GC-TOF-MS) has been widely employed for metabolic profiling of volatiles from highly complex plant matrices. In the past, sample preparation of VOCs analysis relied on different extraction methods, including hydrodistillation (HD), supercritical fluid extraction (SFE), and solvent extractions [66]. These methods are time-consuming; utilize high amounts of solvents, resulting to excessive solvent waste; and causing some temperature-sensitive volatiles to be degraded at high temperatures during extraction [67]. Nowadays, headspace solid-phase microextraction (HS-SPME), a new sample preparation technique, is becoming popular for the screening of volatiles in many plant, food, and beverage matrices [68,69]. HS-SPME is a fast and powerful method which does not require solvents, is easy to automate, and requires only a small sample size [67,70]. Moreover, this method offers the ability to isolate or extract trace multifarious volatile or semi-volatile organic compounds from different product substrates [71]. However, expensive operational costs, including the high cost of standards and chemicals, are the major drawback of this method when compared to low-cost HD and SFE methods. In addition, the absorbent fiber’s application is limited to only some influential high-boiling compounds, such as alcohols [72,73]. On the other hand, GC-TOF-MS can screen hundreds of metabolites and offers higher speed, much greater mass accuracy, and a highly sensitive approach compared to the conventional GC-MS [69,74]. However, it is not capable of quantifying the exact amount of each compound, and is mainly used to explore the overall composition of the target group of compounds [75]. HS-SPME and GC-TOF-MS have been successfully applied as the most modern methods in the metabolomic studies of Zingiberaceae species such as *Curcuma amada* and *Zingiber barbatum* [76,77].

The present study applied the HS-SPME extraction method and GC-TOF-MS detection technique to evaluate the VOC profiles of *K. parviflora* leaves and rhizomes. In addition, the effects of the growing conditions on the volatiles of the tissues were also analyzed to provide a better and more holistic view of the VOC richness of the plant.

## 2. Materials and Methods

### 2.1. Plant Materials

#### 2.1.1. Plant Selection and Preparation

*Kaempferia parviflora* (Accession Number Z1064), from the collection of Tsukuba Plant Innovation Research Center—University of Tsukuba (T-PIRC-UT), Tsukuba, Japan, was used in this study. The morphological features of the plant were noted. *K. parviflora* was reported to have two morphotypes, a red and a green type [78]. The red type was characterized by red-colored leaf margins; red-tinted petioles; and dark, purple-colored rhizome flesh. The green type, on the other hand, had no red-colored leaf margins, no red-tinted petioles, and lighter purple rhizome flesh. It was also mentioned that there were differences in their flavonoid composition. This current study used the green type, as shown in Figure 1.

From a single potted plant, *K. parviflora* plants were propagated into eight individuals planted in separate black plastic pots (21 × 21 cm). The pots contained large granular soil (Akadama Churyu, Kato Sangyo Co., Nishinomiya, Japan) at the bottom and a pre-mixed soil (Hana to Yasai no Bayo Do, Kato Sangyo Co., Nishinomiya, Japan) as the growth medium. The propagated plants were maintained in the growth room of T-PIRC-UT from June 2021 to February 2022 under natural fluorescence light at a temperature of 25 °C and 12-hour light/dark periods. In March 2022, a total of six plants were randomly selected and analyzed for the VOC composition of their rhizomes and leaves.

The unused plant samples were transplanted to separate, larger plastic pots (27 × 28 cm) with the same soil growth media as described above. In May 2022, plants were then moved to the open field of T-PIRC-UT. After six months, the VOC composition of their rhizomes and leaves were analyzed.

#### 2.1.2. Tissue Sample Collection and Preparation

A total of six plants were used for each growth condition: the growth room and the field. For each plant, 2 rhizome and 2 leaf samples were collected. Therefore, a total of 24 samples for each growth condition were analyzed in this study.

Fresh rhizomes and healthy, middle-ranked leaves were collected for the analysis. Rhizomes were rinsed with tap water to remove impurities. Leaf samples were cleaned with 70% ethanol using a soft paper towel. The cleaned samples were kept in ice boxes prior to the next step. Rhizome samples were peeled and sliced into 1.0 cm^2^ pieces using a stainless-steel surgical blade (No. 24, Kai Industries Co., Ltd., Gifu, Japan). The leaves were cut into 2.0 cm tissues using a laboratory scissor, excluding the midrib. The chopped samples were weighed to about 10 g and transferred to a homogenization tube. The tubes were then immediately dipped into liquid nitrogen to freeze the samples inside, then stored in a −80 °C freezer to quench metabolism until further processing.

### 2.2. Sample Processing

#### 2.2.1. Homogenization

The conditions for the homogenization of samples were established following the previously reported methods, with some modifications [76,77]. An MB2000 Multi-bead Shocker (Yasui Kikai Co., Ltd., Osaka, Japan) was used to cryohomogenize the rhizome and leaf samples into fine powder by running into 2 circles at 2800 rpm for 16 s per round. Each fine powder sample, weighing 1.0 g, was placed into a 20 mL headspace screw-top clear GC glass vial (Agilent Technologies Co., Ltd., Waldbronn, Germany). The sample powders were very susceptible to melting, especially the leaf powder. Therefore, as soon as the samples were weighed, GC vials were kept on ice until the all of the chemicals had been added.

#### 2.2.2. Chemicals

All chemicals and reagents used in the study were of analytical grade. The EPA524.2 fortification solution (surrogate standard mixture), *n*-alkane standard solution for determination of the RI (C8–C20), and NaCl were purchased from Sigma-Aldrich Japan (Tokyo, Japan). The EDTA solution for the inhibition of enzymatic activity was supplied by Wako Pure Chemical Industries (Osaka, Japan).

#### 2.2.3. VOCs Extraction

The extraction of leaf and rhizome VOCs was carried out following the previously reported HS-SPME methods, with some modifications [79]. A 50/30 μm DVB/CAR/PDMS preconditioned solid-phase microextraction (SPME) fiber (Supelco, St. Louis, MO, USA) was used in this study.

Sample powders (1.0 g) was placed into the 20 mL headspace GC glass vials (Agilent Technologies Co., Ltd., Waldbronn, Germany). Thereafter, 0.3 g of NaCl (0.3 g/mL), 1.0 mL of 100 mM EDTA (pH 7.5), and 10 μL of an internal standard (IS) of EPA 524.2 fortification solution (20 μg/mL of fluorobenzene, 4-bromofluorobenzene, and 1,2-dichloro-benzene-d4) in methanol were added. Then, the vials were vortexed for 1 min to homogenize the sample powder with the chemicals. VOCs were extracted for 20 min after incubating for 10 min at 80 °C. After the extraction, samples were placed randomly into the CTC PAL autosampler (CTC Analytics AG, Zwingen, Switzerland).

### 2.3. GC-TOF-MS Analysis

The conditions for GC-TOF-MS analysis of VOCs were established following the previously reported methods, with some modifications [76,77]. The collected volatile fractions were injected to an Agilent 6890N (Agilent Technologies, Wilmington, DE, USA) gas chromatograph using the split-less injection mode. The chromatographic column used was Rxi-5Sil MS (30 m × 0.25 mm ID × 0.25 μm; RESTEK, Bellefonte, PA, USA). As a carrier gas, helium (percentage purity > 99.999%) was used at a constant flow rate of 1 mL/min.

The temperature program was initially set at 55 °C for 3 min, then increased at 15 °C/min to 150 °C, then at 3 °C/min to 200 °C, and finally maintained at 200 °C for 2 min. The back-inlet temperature was held at 250 °C. A Pegasus III 4D TOF-MS (LECO, St. Joseph, MI, USA) was used to perform the mass spectral analysis. The TOFMS parameters were set as follows: MS ionization voltage at 70 eV, ionization source temperature at 200 °C, MS scan range (*m*/*z*) at 29–500 amu, and acquisition at 30 spectra/s rate.

### 2.4. Data Processing

The PubChem [80] and ChemSpider [81] chemistry databases were used for the identification of the molecular formula of each annotated peak, the class of each chemical compound, and the CAS-registered number. The proportion of chemical compounds by class in the profile of *K. parviflora* samples was calculated based on the peak intensities of the number of counted annotated compounds in each class, for a total of 100%. The mass spectral peaks where retention indices did not match with those on the reference libraries were described as unknown compounds.

The data processing and provisional VOC identification were performed by following the workflow scheme described by Kusano et al. (2016), with some modifications [82]. The raw data acquired by GC-TOF-MS analysis were exported in a network common data form (NetCDF) file format using Leco ChromaTOF software v4.71.0.0 (LECO, St. Joseph, MI, USA). All data-preprocessing procedures were conducted using the hierarchical multi-curve resolution (H-MCR) method by Jonsson et al. (2006) and MATLAB 7.0 (MathWorks, Natick, MA, USA). The areas of the mass spectral values of the internal standards were calculated by MATLAB R2011b (MathWorks) [83]. The data matrix peaks were then normalized using the method of cross-contribution-compensated multiple standard normalization (CCMN), which eliminates systematic variations and allows for subsequent analysis [71]. The resolved mass spectra generated from the H-MCR method were matched against reference mass spectra from commercially available libraries by applying the NIST mass spectral search program (version 2.2) [84] and the peak annotation-customized software, following the method of Kusano et al. [71]. Reference libraries such as the Wiley’s FFNSC Library (Mass Spectra of Flavors and Fragrances of Natural and Synthetic Compounds, 3rd edition), the Terpenoids Library [85], NIST14 [86], the VocBinBase Library [87], and the Adams Library (4th edition) [88] were used for the identification and estimation of the VOCs. The same or very similar compounds were extracted from the referenced libraries and NIST05 based on the similarity (850 or 900) and the RI difference (<|30 units|) [71]. To determine whether the peaks were from a putatively annotated compound, a similarity of 800, with differences of less than 20 units, was applied when the standard deviation (SD) of the absolute RI difference between these compounds was less than 8.8 units.

### 2.5. Statistical Analysis

The SIMCA 14.0 software (Umetrics AB, Umeå, Sweden) and SPSS 22.0 (SPSS Inc., Chicago, IL, USA) were used for the multivariate analysis. The normalized peak areas of the identified VOCs were log_2_-transformed and then statistically analyzed with the LIMMA package [89]. The significant differences between the identified chemical classes in the HS samples from two tissues and two growth conditions were analyzed through one-way ANOVA (Duncan’s multiple range test).

## 3. Results

### 3.1. The VOC Profile in the Headspace of K. parviflora Rhizomes and Leaves

Using the HS-SPME-GC-TOF-MS-based metabolomics approach, a total of 395 and 426 mass spectra peaks were observed on the VOC profiles of *K. parviflora* samples from the growth room and field conditions, respectively. The obtained MS peaks were provisionally identified by following the methods from Kusano et al. [71]. The interpretations of those spectral matrices were conducted by comparing the retention index of each peak to reference spectra from well-reported libraries [85,86,87,88]. Based on the >95% matching degree of similarity, a total of 85 detected peaks were assigned as putatively annotated compounds out of 395 detected mass spectra in the growth room samples. On the other hand, 100 putative peaks were identified out of 426 matrices in the field samples. The names of the annotated compounds from the samples are listed in Appendix A. The proportion of the main organic compounds was calculated based on the molecular formula of each annotated peak, and their chemical classes were determined. The results showed that the main classes of organic compounds in the HS samples of *K. parviflora* were classified into 5 groups according to their chemical properties: monoterpenoids, sesquiterpenoids, oxygenates, and other hydrocarbons (Figure 2). A group of unknown compounds was exclusively present in the growth room samples and constituted the smallest fraction. The terpenes occupied the largest proportion in all samples from the two growth conditions. Sesquiterpenoids constituted the largest proportion in rhizome samples, followed by monoterpenoids, oxygenates, and hydrocarbons. On the other hand, monoterpenoids were the largest group in leaf samples, followed by sesquiterpenoids, oxygenates, and hydrocarbons.

A Venn diagram was illustrated to visualize the qualitative differences in VOCs between tissues and growth conditions (Figure 3). The number of volatile compounds in the leaf and rhizome samples from the growth room were 78 and 75, respectively, and shared a total of 68 VOCs. Likewise, the leaves and rhizomes from the field contained 98 and 96 VOCs, and a total of 94 were common in both tissues. Among these compounds, 40 volatile components were common between tissues and growth conditions, and are separately listed in Appendix A. It is worth noting that 2, 4, 7, and 10 compounds were detected exclusively from the Field_Leaves, Field_Rhizomes, GR_ Leaves, and GR_ Rhizomes samples, respectively. The identities of these compounds are also listed in Figure 3.

These results indicate that there are significant differences in the contents of volatile compounds in the tissues of the plants grown in different conditions. The 40 compounds commonly identified in both growth conditions and tissues were considered as important compounds which can be used as candidate reference or marker compounds for *K. parviflora*.

### 3.2. Multivariate Data Analysis (MVDA) of Omics Data

#### Principal Component Analysis (PCA)

In metabolomic studies, principal component analysis (PCA) is the primary and most widely used unsupervised multivariate analysis; it allows for the extraction of important information from a complex data set with several intercorrelated dependent variables and can reveal outliers, groups, and trends in metabolome data [90].

A total of 145 putative volatiles obtained from the rhizome and leaf samples from two growth conditions (*n* = 24) were analyzed for PCA modeling. The PCA score plot (Figure 4) clearly shows the separation of the samples into four groups, indicating that PCA is a good model to differentiate *K. parviflora* samples based on tissue type and growth conditions. The two components accounted for 69.6% of the variation in the data set, while PC1 and PC2 explained 49.1% and 20.5% of the total variance, respectively.

It was noted that the growth room samples (GR_Leaf and GR_Rhizome) were clustered closely together in QIII, with negative PC1 and PC2 values, suggesting that they have similar volatile composition profiles. On the other hand, the Field_Leaf and Field_Rhizome samples were found in QII and QIV, respectively. Interestingly, it was noted that all leaf samples (Field_Leaf and GR_Leaf), as well as GR_Rhizome, appeared on the left side of the score plot, while the Field_Rhizome group was positioned on the right side, revealing different compositions of their VOCs. Field_Rhizome samples evidently presented positive PC1 and negative PC2 due to the higher numbers and peak intensities of VOCs compared to the other groups.

Loading plots were also generated from PCA to determine the variables responsible for separating the data, and the results are shown in Appendix A. The loading plots showed corresponding variations of VOCs among the different groups.

### 3.3. Estimated Quantitative VOC Composition of K. parviflora Rhizomes and Leaves

To further study the variation in the relative abundance of VOCs between rhizomes and leaves, the average values of normalized peak areas of the individual volatiles of the samples were log_2_-transformed. The results were presented as the fold-change value of a metabolite concentration as normalized relative to the control—in this case, the rhizome samples. The significant differences in the calculations were assessed with a *p*-value < 0.05. Moreover, the log_2_-transformed data explain the relative abundance of compounds in the leaves, whether higher or lower compared to the rhizomes, and are designated with positive and negative values, respectively.

For easier understanding of the distribution of metabolites across the two tissues in each growth condition, scattered plot-based visualizations were created based on log_2-_ transformed data and a *p*-value < 0.05 (Figure 5). Here, the original *p*-value was transformed to –log 10. The positive side of the log_2_FC values on the *X*-axis explains that the metabolites were significantly increased or higher in the leaves relative to the rhizomes (control), while the negative side shows that the compounds were significantly decreased or lower in the leaves, and consequently increased in the rhizomes. The *Y*-axis describes the significant levels of compounds, and the values are represented as gray and red dots.

Appendix A and Figure 5a present the relative abundance of 85 volatile compounds observed among the 2 types of plant tissue in the growth room conditions. Most of the leaf volatiles had significant differences in their estimated abundance relative to those of the rhizomes (*p*-value < 0.05). A total of 51 VOCs in the rhizomes were significantly higher than those in the leaves. On the other hand, a total of 27 volatiles were found to be higher in the leaves. Only 7 compounds were found to be of similar compositions in the two tissues: α-pinene; bornyl acetate; caryophyllene oxide; and unknown compounds **2**, **3**, **4**, **5**.

As indicated in Appendix A and Figure 5b, out of 100 annotated VOCs from the field samples, 11 compounds in the leaves were found to have similar abundances to the rhizomes (*p*-value > 0.05). These were β-myrcene, caryophyllene, 1,4-dichlorobenzene, 7-endo-ethenyl-bicyclo [4,2,0]-oct-1-ene, methyl palmitate, humulene, carveol, nerolidol, allo-aromadendrene epoxide, caryophyllene oxide, and m-mentha-1(7),8-diene. In addition, it was shown that 21 volatiles were significantly more abundant in the leaves than in the rhizomes. In contrast, 68 compounds were significantly higher in the rhizomes. Overall, it is evident that there is wide variation in the relative abundance of VOCs in *K. parviflora* tissues according to the change in growth environment.

### 3.4. Effect of Growth Conditions on the Composition of Common VOCs of K. parviflora Tissues

As mentioned above, a total of 40 compounds were found to be common between the two tissues and growth conditions. Aside from the qualitative assessment, environment-based changes in the relative abundance of VOCs were evaluated among those compounds. The metabolite compositions in tissues from the field condition relative to the growth room condition (control) were recorded based on log_2_-transformed fold-change values of normalized responses of each annotated compound, and the extent of any significant difference was assessed with a *p* value< 0.05 (Appendix A).

In the case of the rhizomes, the relative abundance of almost all volatiles were significantly higher in the field conditions compared to the growth room, except for three compounds: 2 hexenal, octanal, and hexadecenal. A similar trend was observed in leaf volatiles. Only neophytadiene and borneol showed no significant differences with the change of growth environment. These results suggest that in both tissues, the field condition contributed a significant change in the VOC profile of the plant in terms of quality and relative quantity. This is consistent with the PCA results that the Field_ Rhizome and Field_Leaf groups were clearly separated from the growth room samples.

### 3.5. Identification of VOCs as Basis to Substitute Rhizomes with Leaves

In line with the objective of the study, to evaluate whether leaves can be used as substitutes for rhizomes, field data were prioritized and used to identify the volatiles with similar compositions in both tissues. This is because the previously reported information from SIMCA^®^ and the log_2_-transformed values explained that tissues from the field contain significantly higher numbers and relative abundances of VOCs. Moreover, *K. parviflora* plants are normally cultivated in field conditions for commercial purposes.

Based on the log_2_-transformed values, it was already mentioned that a total of 11 leaf volatiles had similar relative abundances of VOCs with those of the rhizomes in the field conditions (*p*-value > 0.05). These are humulene, caryophyllene, 1,4-dichlorobenzene, m-mentha-1(7),8-diene, carveol, nerolidol, bicyclo [4.2.0]-oct-1-ene, 7-endo-ethenyl-, methyl palmitate, β-myrcene, allo-aromadendrene epoxide, and caryophyllene oxide. Box plot diagrams for these compounds were created based on the normalized peak intensities (Figure 6). However, most of these compounds have low peak intensities of <30 (Figure 6a). Only β-myrcene and caryophyllene have high peak intensities of >100 (Figure 6b). There is a possibility that any of these 11 compounds could be used as a basis to substitute rhizomes with leaves.

## 4. Discussion

In this research, a non-targeted approach coupled with HS-SPME-GC-TOF-MS was applied for the identification of VOCs from the rhizomes and leaves of *K. parviflora* plants maintained in both growth room and field conditions. The identified VOCs in the HS of the two tissues were categorized into five chemical classes. The results showed that sesquiterpenes (C_15_H_24_) were the major group in rhizomes. This result was consistent with the results of other studies on *K. parviflora* rhizomes [15,48,49,50]. In other Kaempferia species, such as *K. galanga*, phenylpropanoids constitute the largest proportion of their rhizomes [8,91]. On the other hand, monoterpenes (C_10_H_16_) are the dominant compounds in *K. parviflora* leaves, which is first time that this has been reported for the species. Rhizomes are grown underground and exposed to pathogenic attacks by fungi and bacteria. Sesquiterpenes are known to have antifungal and antibacterial properties [11,92,93]. On the other hand, leaves can be found above the ground, and are more exposed to insects. Monoterpenes are produced to deter insects. These two groups of VOCs exhibit a wide range of biological activities, including antioxidant, anti-inflammatory, anti-tumor, anti-malaria, and anti-neoplastic properties. They are also used for commercial purposes, such as in perfumery, aromatherapy, and food flavoring. Our findings affirm that *K. parviflora* rhizomes and leaves are both potential sources of several active pharmacological compounds with potential commercial applications.

PCA was performed to investigate the relationships between the VOCs in two tissues of *K. parviflora* plants grown in two different environments (Figure 4). The PCA results demonstrated clear separation of the samples into four groups. The two tissues of the growth room samples were closely clustered together, indicating similar VOC profiles. This could be due to the constant light and temperature in the growth room and the plants’ lower exposure to stress. On the other hand, the field samples were separated from growth room samples, possibly due to changes in the VOCs’ composition and abundance. Evidence from the literature supports the finding that emissions of terpenes from plants are known to be both light- and temperature-dependent [94,95,96,97]. Competitive interactions between plants, pollinator attraction, and plant defense processes stimulate terpene production [95]. These conditions were possibly more present in the uncontrolled field conditions than the growth room conditions.

A significant separation of rhizome and leaf samples from the field was also observed on the PCA plot. Rhizomes are storage organs with large secretory tissues, which can explain their more diverse VOC composition and higher abundance in rhizomes than in leaves. In addition, individual samples of the two tissues showed that they were slightly scattered from each other, while those of the growth room plants were closely clustered together. This can be attributed to the different responses of the individual plants to the environment in an open field than in a controlled growth room environment.

The constructed Venn diagram was used to visualize the distribution and number of identified VOCs in all samples. It was shown that there was not much difference in the identified numbers of VOCs between the two tissues in both growth conditions. However, the PCA analysis showed better separation of the samples into four groups. This is because the PCA results account not only for the number of identified VOCs, but also for their relative abundance.

Log_2_-transformed data shown in Appendix A also confirmed the significant differences in the relative abundance of VOCs between the two tissues. Furthermore, in PCA, the separation was more obvious between the tissue samples from the field and those from growth room group even though they share small number of differences in detected VOCs, such as 4 and 6, respectively. Finally, the results of PCA and the Venn diagram collectively explained the greater influence of the environment on the relative abundance of volatiles than their qualitative differences.

A total of 78 and 98 annotated VOCs were detected from the HS of rhizome samples of *K. parviflora* from the growth room and the field, respectively. Previous studies of *K. parviflora* rhizome oil reported 21–32 volatile compounds, depending on the variety and collection site [15,48,49,50]. A noteworthy observation was the higher number of compounds detected in this current study. This may be due to the differences in the extraction methods and analytical tools used. Some temperature-sensitive volatiles can be degraded when high temperatures are used during the analysis, which is eliminated in the HS-SPME. Adam et al. (2022) also confirmed that more volatile components were detected using HS-SPME than hydro-distillation in *Mentha* species [66]. Moreover, semi-volatiles, such as oxygenates and some hydrocarbons, were also detected in this study, but had not been reported in previous studies. These findings clearly showed the advantage of the HS-SPME-GC-TOF-MS approach in analyzing VOCs of plants.

Based on previous reports, a total of 54 volatile compounds were found to be present in the rhizomes of field-grown *K. parviflora,* and only 11 of those VOCs were reported to be common among them [15,48,49,50,51]. In the current study on rhizome volatiles of field-grown plants, these 11 VOCs were also detected. In addition, 48 other VOCs were newly reported here, while 30 compounds were not observed in this study. These could also be attributed to the detection techniques used, as well as the differences in the genetic backgrounds of the plant sources and geo-climatic conditions [4].

The VOC profiles of *K. parviflora* leaves were also explored in this study for the first time using the HS-SPME-GC-TOF-MS approach. A total of 75 and 96 VOCs were detected from the growth room and field leaf samples, respectively. Among these, only 40 compounds were detected to be common in both conditions (Figure 3). The log_2_-transformed data showed that the relative abundance of these 40 VOCs was significantly higher in the field samples than the growth room samples (Appendix A). Similar results were observed in rhizomes. The differences in relative abundance between the two growth conditions could be due to the presence of biotic and abiotic factors. Plants in the field are more exposed to increased daylight intensities and temperatures, contributing to increased terpene biosynthesis and emissions by regulating the activity of synthase enzymes [95,96,97]. In addition, plants grown in the field are also more exposed to various kinds of biotic stresses, such as insect and herbivore attacks, and the plants use VOCs to mitigate those stressors.

It is interesting to note that there is a high level of similarity between the two tissues in terms of their identified VOCs (Figure 3). In both growth conditions, high numbers of VOCs were shared between the rhizomes and leaves. A total of 68 out of 85 VOCs in growth room samples and 94 out of 100 VOCs in the field samples were detected to be shared by the two tissues. These data show that the volatiles in leaves, in terms of composition, are comparable to that of the rhizomes. Thus, *K. parviflora* leaves can be a potential alternative source for the industrially important volatiles of rhizomes.

However, the main difference in the VOCs of the tissues lies in the relative abundance of these compounds. This is estimated in the study by calculating the log_2_-transformed values from the GS-MS-annotated peaks of the compounds (Appendix A). The results showed that the relative abundances of most of the rhizome volatiles were significantly higher than those of the leaves (*p*-value > 0.05). This may be related to the large VOC pool synthesized in the secretory tissues of rhizomes. VOCs are synthesized in oil and resin ducts, or glandular trichomes, and easily released from storage pools due to high temperature or herbivore feeding or movements on surface [94]. In the case of *K. parviflora,* the rhizome has many oil ducts, while the leaves have few and no trichomes. On the contrary, there were also VOCs which were significantly abundant in the leaves compared to the rhizomes (*p*-value < 0.05). These compounds have the potential to substitute the same compounds from rhizomes, while also taking into consideration their economic values.

## 5. Conclusions

The VOC profiles of *K. parviflora* rhizomes and leaves collected from plants grown in the growth room and open field were explored for the first time using the HS-SPME approach coupled with non-targeted GC-TOF-MS. This study affirmed that the utilized technique used is more efficient for analyzing the VOCs of the samples. It also confirmed that the composition of VOCs is highly affected by the growth conditions. The field-grown plants showed higher numbers and abundances of volatiles compared to plants grown in the growth room. In terms of the VOC composition of the tissues, monoterpenes were found to be the major VOCs of leaves, whereas the rhizomes were rich in sesquiterpenes. The majority of the VOCs present in the rhizomes were also found in the leaves. However, the concentrations of most of the rhizome VOCs were higher than those of the leaves. Nevertheless, the findings confirmed that *K. parviflora* leaves are also a rich reservoir of volatile organic compounds. Therefore, for economic purposes, fresh rhizomes of field-grown plants, as well as their leaves, can be utilized as sources of valuable volatiles.

## Figures and Tables

**Figure 1 metabolites-13-00651-f001:**
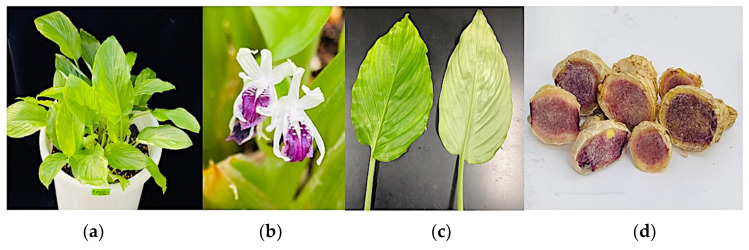
Habitat (**a**); flower (**b**); leaf (**c**); and rhizome (**d**) of the *Kaempferia parviflora* accession investigated in the study.

**Figure 2 metabolites-13-00651-f002:**
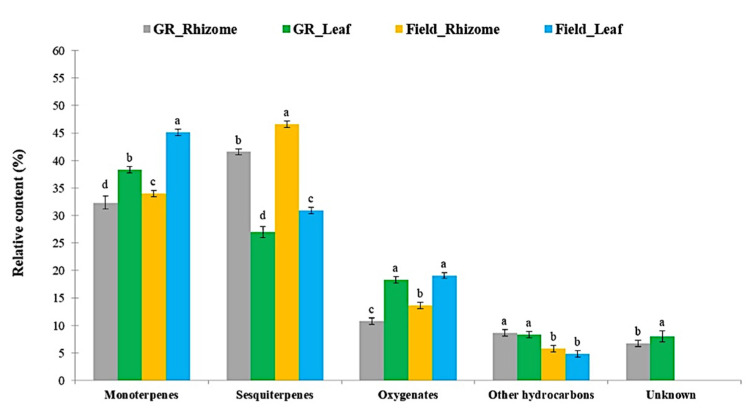
Relative content of identified classes of organic compounds in the headspace (HS) samples of *K. parviflora* rhizomes and leaves from growth room and field conditions. Different lowercase letters in the same volatile categories indicate significant differences at the *p* < 0.05 level.

**Figure 3 metabolites-13-00651-f003:**
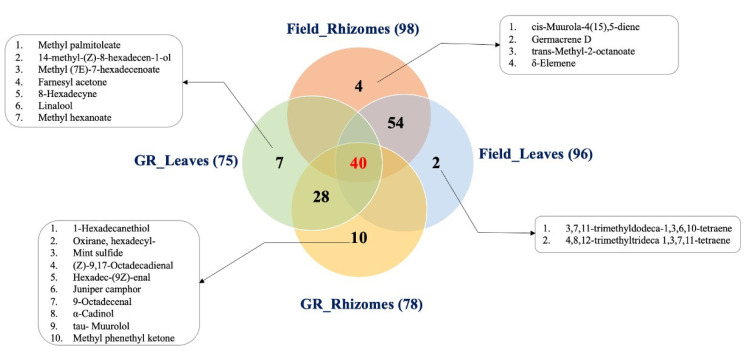
Venn diagram of volatile components found in *K. parviflora* rhizomes and leaves from field and growth room (GR) conditions.

**Figure 4 metabolites-13-00651-f004:**
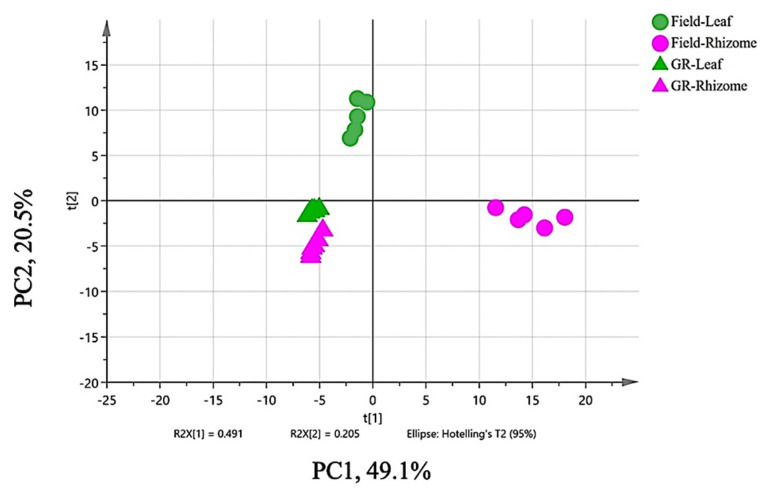
The PCA score plot generated from the GC-TOF-MS data of *K. parviflora* rhizomes and leaves from field and growth room (GR) conditions. The first two principal components (PCA) accounted for a total of 69.6% of the variance. (*n* = 24).

**Figure 5 metabolites-13-00651-f005:**
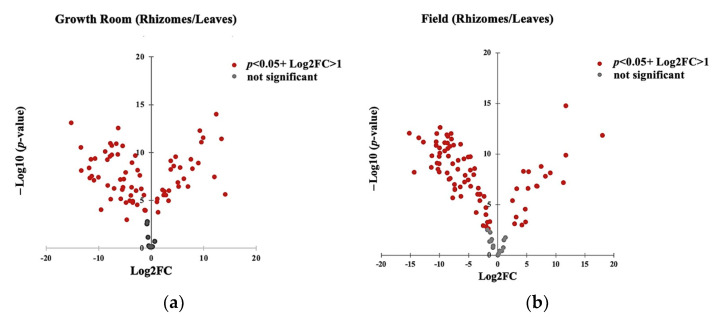
Scatter plot generated from log_2_-transformed values of normalized peak intensities of annotated compounds identified from the rhizomes and leaves of *K. parviflora* grown in growth room (**a**) and field (**b**) conditions. The *X*-axis represents the log_2_FC values and the *Y*-axis represents the –log10 of the *p* value. The red dots of the positive side explain the significantly increased volatiles of the leaves, and negative side represents the significantly increased volatiles of the rhizome (*p* value < 0.05). Grey dots represent no significant difference in relative abundance.

**Figure 6 metabolites-13-00651-f006:**
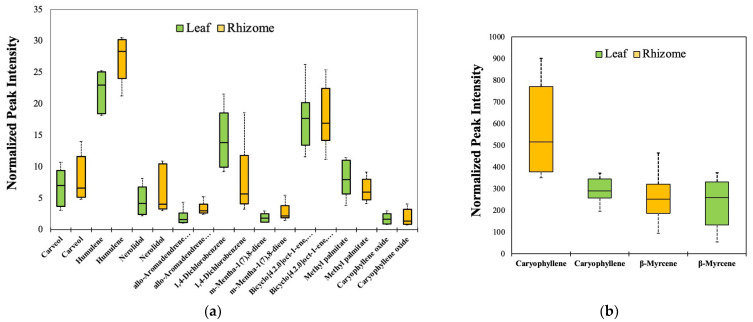
Boxplot diagrams for normalized peak intensities of volatile compounds identified from the rhizomes and leaves of *K. parviflora* grown in the field, with no significant differences (*p*-value > 0.05) among their relative abundance. Nine VOCs with peak intensities of <100 (**a**). Two VOCs with peak intensities of >100 in both tissues (**b**). *Y*-axes are peak intensity values from GC-TOF-MS. The mean is marked with a horizontal line; the bottom and the top of the box represent the 25th and 75th percentiles; and the whiskers represent the minimum and maximum values.

## Data Availability

The data presented in this study are available in article and Appendix A.

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
