# Peer review of "Exploring Volatile Organic Compounds in Rhizomes and Leaves of Kaempferia parviflora Wall. Ex Baker Using HS-SPME and GC–TOF/MS Combined with Multivariate Analysis"

_metabolites, 2023, doi:10.3390/metabo13050651_

Round 1

Reviewer 1 Report

The manuscript is covering the topic about determination of some compounds in different parts of plant Kaempferia parviflora with the use of head-space analysis followed by TOF/MS. In general, the manuscript is well written and after some modifications and comments will be suitable for publication in Metabolites journal.

Comments:

In abstract, all abbreviations must be explained regardless their use in main body text

Introduction should focus more on techniques on determination, hence the need to improve this part of manuscript while providing why the presented method has the novelty. Also the limitation must be provided, here or in the conclusion section. More details (at least one sentence on sample preparation conditions should be introduced).

Figure 1 should be replaced by illustration, pictures are less informative.

The results and data analysis were done correctly. However, what I miss is the sample preparation optimization procedure while I realize the sample prep was adapted, hence some comments on conditions choice must be provided. Chromatograms obtained during analysis showing blank, sample or at least one chromatogram with some identified compounds for comparison should be provided. Also, how the condition of HS-SPME and GC were optimized – this is crucial in determination of unknown compounds.

English is fine. Minor corrections should be done by native speaker.

Reviewer 2 Report

This study compared the chemical composition of VOCs in the leaves and rhizomes of Kaempferia parviflora, a ginger plant in the Southeast Asia. The manuscript contains some useful information on the chemical differences caused by growth environment, but have defects in the rigor of chemical analysis and data analysis.

---One clear defect is lack of quantitative information on the yields of VOCs in the leaf and rhizome samples as well as the absolute concentrations of identified compounds. The authors used the "concentration" word in many places in the text, but they are just relative abundances of MS signals, not real concentrations. Without these quantitative information (such as % VOCs in each material and the concentrations of selective bioactive compounds with extracted VOCs), it is hard to judge the pros and cons of different materials and cultivation conditions. The authors should make some experimental efforts to address it in the revision.

---It is puzzling that GR-leaf and GR-Rhizome have comparable chemical compositions as shown in Figure 4 considering they are completely different tissues. In addition, this is inconsistent to the Venn diagram in Figure 3, in which 17 (7+10) compounds differ between GR-leaf and GR-Rhizome while only 4 (2+2) compounds differ between Field-leaf and Field-Rhizome). The discussion on this phenomenon in the Discussion (line 471-476) is quite weak and not helpful. Please provide a reasonable explanation.

---For multivariate data analysis, if the unsupervised PCA can achieve clear sample separation as shown in Figure 4, then supervised OPLS-DA is not needed in practice as it is more biased than PCA.

--- The Results section should not contain the contents that belong to the Materials and Methods (such as Line 249-256, 359-366, and other places)) and the Discussion (the sentences with the references for discussions). Please revise.

---Don't capitalize the compound name in the sentences, such as line 480 and other places.

Reviewer 3 Report

Thank you for giving me the opportunity to review the manuscript by  May San Thawtar  et al., entitled “Exploring volatile organic compounds in rhizomes and leaves of Kaempferia 2 parviflora Wall. ex Baker Using HS-SPME and GC–TOF/MS combined with 3 multivariate analysis", submitted for publication in Journal of metabolites.

The presented work deals HS-SPME with GC-TOF-MS was used to investigate the VOC composition in the leaves and rhizomes of K. parviflora plants 23 that were cultivated in growth rooms and the field.

The manuscript's writing is usually bad, although I did spot a few small grammatical and syntax mistakes, as well as capitalization and punctuation issues scattered throughout the text.

Before making a final choice on the work, you should take into account the numerous in-depth comments, critiques, concerns, and ideas I've provided below. I think the rewritten document will produce a considerably different form from the one it is in now, taking into account the criticisms I've provided below. In light of the fact that this work typically yields some intriguing results, I advise you to submit it after modifying it.

Sincerely

The main criticism points are:

1-      There are many grammatical, punctuation, syntax errors, so sever English language editing is needed. For example:

-          Line #20 – use (for their) instead of (its).

-          Line #26 – use (detected in )   instead (detected from).

-          Line #28 – (sesquiterpenes were dominant) not (sesquiterpenes )

-          Line #29 – (a significantly) not (significantly

-          Line #30 – ( a High) not (High)

-          Line #32 – ( lies in ) not (on)

-          Line #34 – (can be further) not (can further be)

Also( used as an) not (used as)

-          Line #47 – ( The major classes) not (Major classes)

-          Line #48 – (most aromatic plants) not (most of aromatic plants)

-          Line #50 – add (and) before (esters)

-          Line #50 – (derivates) replace with (derivatives)

And so on

-           There are many spelling errors Please review and amend it .

2-       Material and method Lacks recent references and VOCs extraction references not mentioned

3-      What is the experimental design? its not clear

-

Round 2

Reviewer 2 Report

Extensive editing in writing was conducted in this revision. However, some of my comments are not well addressed.

---The authors stated that "GC-TOF-MS is not able to quantify or calculate the concentrations of each and individual compounds". However, I did not ask for the identification or quantification of each compounds. I wonder any of these terpenoid compounds have been confirmed by their authentic standards. If so, please state it the paper and conduct the quantitative analysis if possible. This will greatly improve the quality of this study.

---The  following sentences in Results should be placed into the Methods section:

"The obtained MS peaks were provi-751 sionally identified following the methods from Kusano et al. [71]. The interpretation for 752 those spectral matrices was conducted by comparing the retention index of each peak to 753 reference spectra from well-reported libraries [83–86]."

"The PubChem [88] and ChemSpider [89] chemistry databases were used for the 760 identification of the molecular formula of each annotated peak, the class of each chemical 761 compound, and the CAS registered number, to determine the class and calculate the 762 proportion of the main organic compounds. The proportion of chemical compounds by 763 class in the profile of K. parviflora samples was calculated based on peak intensities of the 764 number of counted annotated compounds in each class as a total of 100%. The mass 765 spectral peaks where retention indices did not match with those on the reference libraries 766 were described as unknown compounds."

N/A

Round 3

Reviewer 2 Report

The revisions have addressed my comments.

A careful proofreading should be performed.